# Examining the mental health outcomes of school-based peer-led interventions on young people: A scoping review of range and a systematic review of effectiveness

**Thomas King** ✱*, **Mina Fazel**

Department of Psychiatry, Warneford Hospital, University of Oxford, Oxford, United Kingdom

* Thomas.king@psych.ox.ac.uk

**Data Availability Statement:** All relevant data are within the paper and its Supporting Information files.

## Abstract

Schools worldwide have implemented many different peer-led interventions with mixed results, but the evidence base on their effectiveness as mental health interventions remains limited. This study combines a scoping review and systematic review to map the variations of peer-led interventions in schools and to evaluate the quality of the existing evidence base. This scoping review and systematic review evaluated the existing literature across 11 academic databases. Studies were included if they reported a peer-led intervention that aimed to address a mental health or wellbeing issue using a peer from the same school setting. Data were extracted from published and unpublished reports and presented as a narrative synthesis. 54 studies met eligibility criteria for the scoping review, showing that peer-led interventions have been used to address a range of mental health and wellbeing issues globally. 11 studies met eligibility criteria for the systematic review with a total of 2,239 participants eligible for analysis (929 peer leaders; 1,310 peer recipients). Two studies out of seven that looked at peer leaders showed significant improvements in self-esteem and social stress, with one study showing an increase in guilt. Two studies out of five that looked at peer recipient outcomes showed significant improvements in self-confidence and in a quality of life measure, with one study showing an increase in learning stress and a decrease in overall mental health scores. The findings from these reviews show that despite widespread use of peer-led interventions, the evidence base for mental health outcomes is sparse. There appear to be better documented benefits of participation for those who are chosen and trained to be a peer leader, than for recipients. However, the small number of included studies means any conclusions about effectiveness are tentative.

## Introduction

Addressing the mental health needs of school-aged children and adolescents is a global priority [1] with one in eight young people in the United Kingdom experiencing a diagnosable psychiatric disorder [2]. Schools are now widely recognised as an important setting for early mental

**Funding:** The authors received no specific funding for this work.

**Competing interests:** The authors have declared that no competing interests exist.

health intervention [1, 3]. This has contributed to a rise in the use of peer-led interventions to address mental health needs, a method that is used in schools across the globe, both in low and high resource settings with practice across Europe, Africa, Asia and North America. An estimated 62% of schools in England have, in a national school survey, stated that they offer peer-led intervention [4]. Peer-led interventions take a variety of forms and names, such as peer mentoring, peer buddying, peer counselling and peer education [4, 5]; in this paper we will use the term 'peer-led' to include all of these activities, and 'peer leaders' or 'peer recipients' (whether giving or receiving an intervention, respectively) to describe participants in these interventions.

Peer-led interventions typically involve the selection, training and supervision of a group of pupils in preparation for a supportive or educational role among similar-aged pupils in their school [6]. In preparation, peer leaders are typically taught basic counselling and communication skills, which are seen as key skills for the role [7]. Previously, young people have been trained to offer support not only for mental health, but across a range of school-specific areas, such as school transition, isolation, and bullying [5]. They have also delivered physical health promotion interventions, aiming to increase physical exercise [8], healthy eating [8, 9], and smoking cessation [9, 10] among their student body. However, despite this apparently wide use, the evidence base for peer-led interventions to address mental health outcomes remains limited.

Therefore, this review will focus on the use of peer-led interventions to address mental health and well-being in schools. Peer-led programmes have been used in isolation but are more commonly used alongside other services, such as school counselling, to address lower-intensity needs and provide simple psychosocial support. This has the potential to allow staff members and professional in-school services to focus on higher level issues [11]. The interventions are typically offered on a 1:1 or group basis. Some have a set programme whilst others encourage users to arrange appointments or offer a more informal 'drop-in' service. Many programmes are made available to the whole school, while others are targeted at specific populations, such as victims of bullying [12].

There are many compelling reasons why peer-led interventions are popular in the school setting. Firstly, these programmes are relatively resource-light and may therefore be more acceptable and feasible to run in schools. Schools can often provide a range of potential delivery locations, which may benefit the sometimes ad hoc nature of the peer-led format. For instance, previous peer-led interventions have taken place in classrooms, playgrounds, common room areas, after-school clubs and dining rooms. This flexibility is an important consideration for many educational settings. The low resource requirement also lends itself to scalability, enabling schools to increase the reach of these interventions if needed. Furthermore, schools have a large pool of students from which to select their peer leaders who are often keen to play this role and contribute to their school community, with a possible endorsement as good school citizens.

Secondly, school-aged children have previously been used to address a range of separate but potentially interconnected problems affecting their peers, such as school connectedness [5, 13], communication and social skills [14, 15], and support with school transitions [16]. The use of peer leaders to support victims of bullying has also been widely studied, with mixed results [12, 17–19]. Furthermore, peer leaders have been used to help those with chronic health conditions [20] and also encourage healthy living behaviours [21, 22]. The relatively widespread use of peer-led interventions across disciplines, and the absence of any synthesis of peer-led interventions that target mental health outcomes, catalysed this review.

Lastly, the case for the peer-led approach in schools is strengthened by the increasingly appreciated importance of social influence and peer attachments in the adolescent years [23],

combined with evidence showing that young people more commonly turn to informal sources of support, including friends, for psychological needs [2]. This may subsequently lead young people to be more inclined to seek a similar-aged peer for issues around their mental health and wellbeing.

Despite widespread use of peer-led programmes suggesting their acceptability in a school environment, there are still a number of barriers to consider. These relate both to the implementation of these programmes and the personal effects on its users. Firstly, the increase in social awareness during adolescence may also act as a deterrent to confiding in a peer leader if the young person fears judgement, ridicule or even rejection from their allocated peer. Secondly, some studies have encountered low programme usage rates, either due to poor awareness of the programme [24], not believing that a peer leader would be able to help them [25], or preferring to seek other sources of support, such as one's existing friends [18]. One study identified capacity and resourcing issues, as well as lack of interest from students, as barriers to maintaining a peer-led programme [11].

UK governmental reviews have pointed to the need for further research into peer-led programmes. For example, the 'Future in Mind' [26] report identified the integration of mental health support into schools as a priority, with a particular emphasis on the development of peer support. This culminated in the production of a research review in 2017 which identified some areas of current research around the development and efficacy of peer-led programmes in the UK [27]. A key finding of the review was that the evidence base is sparse and lacks overall quality.

As we were not able to identify any appropriate systematic or scoping review of the breadth and quality of peer-led programmes in schools and their mental health effects, this review was conducted with the specific aims to:

1. Conduct a scoping review of the range of peer-led interventions used to address mental health outcomes in schools.

2. Conduct a systematic review to collate and evaluate the data on the effectiveness of school-based peer-led interventions on mental health outcomes.

3. Map the range of mental health outcomes that have been identified.

## Materials and methods

### Search strategy and selection criteria

The possible sources of eligible studies are broad, given the potential physical, mental and public health, and educational focus of interventions. Therefore, we searched the following 11 electronic databases for eligible studies: PsycINFO; PubMed; EMBASE; CINAHL; CENTRAL; BEI; Scopus; Web of Science; ERIC; Social Sciences Citation Index (SSCI); and Social Care Index. The list of search terms [see S1 Appendix for one database search] was developed after an initial scan of the literature and using online database thesaurus tools. Once the search terms had been compiled, pilot searches were run to ensure that key texts were appearing in the search, especially given the different terms used for these activities and different ways in which they have been evaluated. Any search terms that did not appear to be returning any relevant results were removed from the search. Search strategies such as truncations, e.g. 'psych*', and MeSH terms were employed. The searches were kept as similar as possible between databases. We did a systematic search of studies published in any language up until 20th December 2018 initially, with a repeated search up until 12th May, 2020. No earliest publication date was applied. No restrictions were placed on publication date, country or language. The broad

search categories included mental health, schools, and peer-led interventions and included up to 120 search terms. Studies were identified with a wide range of synonymous search terms for 'mental health', 'peer support', 'adolescent', 'school', and 'intervention'. Programmes with peer leaders from outside the school, whether from the community, another school or a university, were not included. Any intervention that primarily used an adult facilitator to lead and actively guide peer-to-peer discussions were also not eligible.

One reviewer screened titles and abstracts for relevance for both the scoping review and systematic review, and then screened the remaining full texts (see Figs 1 and 2). Any that were unclear were brought to the second reviewer for discussion. For both reviews, sources were screened against the respective eligibility criteria and relevant data were extracted from included studies. Study authors were contacted where any further information or details were needed. Forward and backward referencing was performed on all included and any relevant studies. A range of grey literature sources were searched, including conference proceedings, dissertations, and government documents. The protocol for the systematic review is available online [28].

## Role of the funding source

There was no funding source for this study.

## Scoping review

### Inclusion criteria

The scoping review included peer-led interventions targeting mental health or wellbeing outcomes. The interventions must have taken place within a primary or secondary school and have been predominantly led by students within that school. As definitions of 'mental health' and 'wellbeing' can vary, the studies included were those that had identifiable mental health outcomes. Any programmes with an online element were included as long as they were peer-led and based within the school. There were no restrictions based on research design or quality for the scoping review [29].

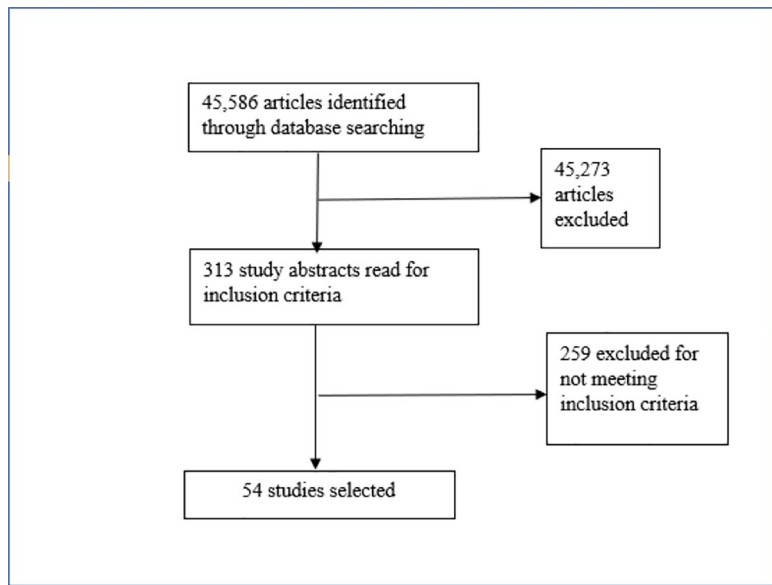

**Fig 1. PRISMA flow diagram for scoping review.**

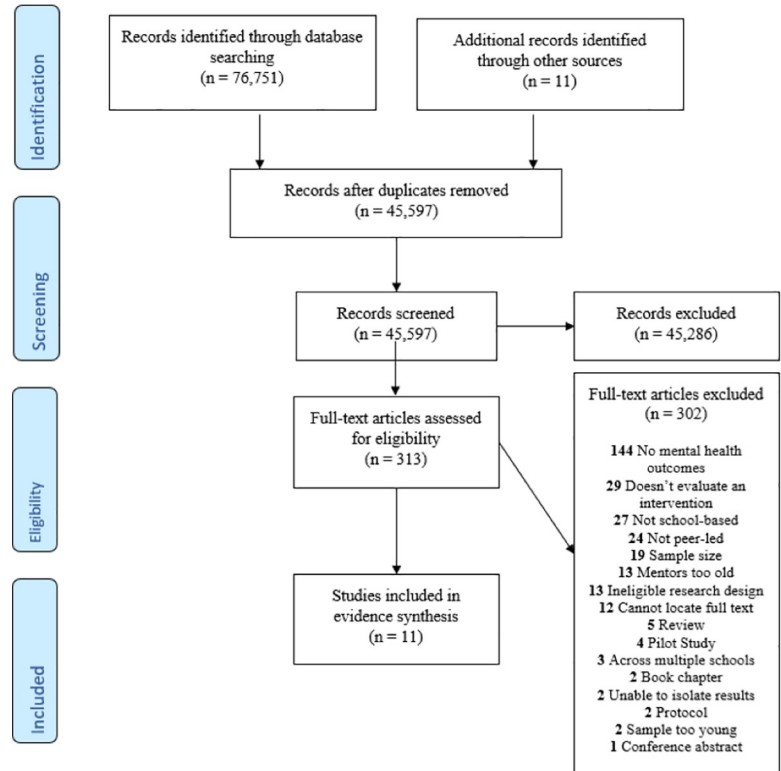

**Fig 2. PRISMA 2009 flow diagram for systematic review.** *From*: Moher D, Liberati A, Tetzlaff J, Altman DG, The PRISMA Group (2009). *Preferred Reporting Items for Systematic Reviews and Meta-Analyses*: The PRISMA Statement. PLoS Med 6(7): e1000097. doi:10.1371/journal.pmed1000097.

## Results

The results of the scoping review are in Table 1. A total of 54 studies are included that show the range of peer-led interventions across the globe.

The interventions included those to support positive behaviours and health (buddy benches; wellbeing focus; connection focus) as well as targeting higher risk populations such as those with suicidal thoughts. They often involved training the peer leader to conduct a further educational or training/workshop intervention for their peers. A number of interventions addressed the mental health impact of specific experiences, such as bullying and school transitions, while others aimed to improve mental health in order to prevent certain negative outcomes, such as school dropout. Of the included studies, 46 out of 54 were conducted in high-income countries, of which half were in North America. In total, the included interventions took place mostly in secondary schools (89.1%) with approximately one fifth also or exclusively in primary schools (19.6%).

## Systematic review

### Methodology

We followed the PICO (Population, Intervention, Comparator and Outcome) format to develop our research question [78]. We completed the systematic review in accordance with the 2009 PRISMA statement [28, 79] and registered it with PROSPERO (CRD42018116243).

**Table 1. School-based peer-led interventions identified in scoping review.**

| Author and year of publication | Country | Primary (P) or Secondary (S) | Summary |
|---|---|---|---|
| **Bullying** | | | |
| **Cowie (1998) [30]** | England | S | 5 x peer counselling anti-bullying programs: all in secondary schools; 4 included 1:1 and group sessions; 1 telephone help-line; took place in designated rooms. |
| **Cowie and Olafsson (2000) [6]** | UK | S | Peer mentoring for bully victims |
| **Hurst (2001) [31]** | England | S | Year 8 students offering ad hoc support to bully victims at lunchtimes |
| **Price and Jones (2001) [32]** | Wales | S | Year 11 students as peer counsellors for year 7 bully victims–informal sessions |
| **Lines (2005) [33]** | UK | S | Year 10 girls and Year 9 boys trained to support bully victims in informal sessions |
| **Hutson and Cowie (2007) [34]** | UK | S | E-mail peer support group in all-boys school |
| **McElearney et al. (2008) [35]** | Northern Ireland | P | Befriending programme where 10 and 11 year olds support 5–8 year olds to reduce the impact of bullying |
| **Houlston and Smith (2009) [36]** | UK | S | Year 10 students as peer counsellors for bully victims |
| **Roach (2014)** | England | P + S | 1:1 and group-based support for victims of bullying |
| **Suicide Prevention** | | | |
| **Tse et al. (1994) [37]** | Hong Kong | S | 'HIT-MAN' Suicide prevention programme; trained to identify signs of suicidality in peers |
| **Barber et al. (1995) [38]** | USA | S | One-off workshop around stress and suicide delivered by high school psychology seniors to sophomores (two years below) |
| **Wyman et al. (2010) [39]** | USA | S | Schoolwide messaging as part of suicide prevention scheme |
| **Calear et al. (2016) [40]** | Australia | S | Peer leaders spreading positive messages in school as suicide prevention initiative |
| **Wright-Berryman et al. (2018) [41]** | USA and Canada | S | Students become part of 'Hope Squads', centred on referral of suicidal peers to appropriate adults |
| **Zachariah (2018) [42]** | India | S | Mindfulness-based suicide prevention program |
| **Depression** | | | |
| **Connor (1995) [43]** | USA | S | Peer support groups addressing range of problems run by 2 student peer supporters at a high school |
| **Parikh et al. (2018) [44]** | USA | S | High school students trained to design and implement peer-to-peer depression awareness campaigns |
| **Substance Misuse** | | | |
| **Winters and Malione (1975) [45]** | USA | S | 2 programs: 1) Student-run hotline providing information and referrals; 2) Drug, alcohol and tobacco education |
| **Karaca et al. (2018) [46]** | Turkey | S | Substance misuse education sessions |
| **Facilitating Peer Social Connections** | | | |
| **Abu-Rasain et al. (1999) [47]** | Saudi Arabia | S | Peer support by same-aged students in boys' secondary school to address loneliness |
| **Karcher (2005) [13]** | USA | S | 'Developmental' mentoring by older students in a high school to promote connectedness |
| **Gallacher (2011) [48]** | Scotland | P | 'Playground Pals' initiative–primary school children trained to encourage happy and positive playground environment |
| **Freed and Lowenstein (2017) [49]** | USA | S | 'AHA! Peace Builders' programme–high school students conduct 'Connection Circles' with peers |
| **Griffin et al. (2017) [50]** | USA | P | Playground 'Buddy Bench' in Elementary School |
| **Psychoeducation** | | | |
| **Corn et al. (1992) [51]** | USA | S | 1:1 and group sessions for psychoeducation and emotional support |
| **O'Hara (2011) [52]** | UK | P | Year 9 students paired with Year 7 students to improve emotional literacy |
| **O'Reilly et al. (2016) [53]** | Ireland | S | Training post-primary aged children to deliver mental health education workshops |
| **School Transition** | | | |
| **Slater et al. (2004) [54]** | England | S | 1:1 and group interventions by older students to support year 7 students with secondary school transition |
| **Brady (2014) [55]** | Ireland | S | Older students offer 1:1 support to support younger students with school transition |
| **Pandina (2015) [56]** | USA | S | Older students support incoming high school students to prevent onset of harmful behaviours |

*(Continued)*

**Table 1.** (Continued)

| Author and year of publication | Country | Primary (P) or Secondary (S) | Summary |
|---|---|---|---|
| **Lorenzo (2019) [57]** | USA | S | Peer mentoring to ease school transition |
| **Body Image** | | | |
| **Stock et al. (2007) [58]** | Canada | P | 4th-7th grade students teaching K-3rd grade students in 1:1 pairs about healthy living, including body image |
| **Ishak et al. (2019) [59]** | Malaysia | S | Health education program to promote body positivity |
| **Academic Issues** | | | |
| **Channon et al. (2013) [60]** | Wales | S | Year 9 mentors for year 7 peers to support wellbeing and academic achievement |
| **Johnston-Wilder et al. (2015) [61]** | UK | S | Peer-to-peer counselling for maths anxiety |
| **General Mental Health Support** | | | |
| **Hamburg (1972) [62]** | USA | S | High school and junior school students help others with personal problems and stress |
| **Buck (1977) [63]** | USA | S | Cross-age peer counselling for range of emotional and behavioural problems in high school |
| **Armstrong et al. (1987) [64]** | Canada | P | Self-esteem effects of 1:1 buddying with physically disabled peers |
| **Philip-Moustakas (1994) [65]** | USA | S | SPARK student program–High school students provide counselling and refer to other help sources |
| **Froh (2004) [66]** | USA | S | Peer supporters in grades 6–12 trained in 'Natural Helpers' program |
| **Bradley (2016) [67]** | England | S | Peer mentoring groups to improve self-esteem in secondary school students with Autism |
| **Warner (2018) [68]** | UK | S | Peer coaching to increase self-esteem and reduce test anxiety |
| **Maree (2018) [69]** | South Africa | S | Peer counselling to enhance sense of self |
| **Day, Campbell-Jack and Bertolotto (2020)** | UK | P + S | Wide range of peer support interventions with general aim of improving mental health and wellbeing |
| **Mental Health Promotion** | | | |
| **Holsen et al. (2015) [70]** | Norway | S | Universal intervention co-led by older peer leaders to promote positive mental health |
| **Transition to Adolescence** | | | |
| **Ellis et al. (2009) [5]** | Australia | S | Older students supporting younger students in transition to adolescence |
| **Referrals to Professional Services** | | | |
| **Winters and Malione (1975) [45]** | USA | S | 2 programs: 1) Student-run hotline providing information and referrals; 2) Drug, alcohol and tobacco education |
| **Sexual Health and Wellbeing** | | | |
| **Mason-Jones et al. (2013) [71]** | South Africa | S | Peer education program targeting sexual health and wellbeing |
| **Healthy Lifestyle** | | | |
| **Ronsley et al. (2013) [72]** | Canada | P + S | Older students teaching younger students about eating and living healthily |
| **Behavioural Problems** | | | |
| **Lazerson (1980) [73]** | USA | P + S | Older students with current behavioural problems giving brief daily learning sessions to younger students with similar problems |
| **Tobias and Myrick (1999) [74]** | USA | S | 17–18 year old students counselling 11–12 year old students with problem behaviours |
| **School Dropout Prevention** | | | |
| **Nenortas (1987) [75]** | USA | S | Peer counselling groups to increase self-esteem and reduce absenteeism and dropout |
| **Physical Health Education** | | | |
| **Diao et al. (2020) [76]** | China | P + S | Physical health education program to improve Quality of Life outcomes; range of health promotion strategies |
| **Widayanti (2018) [77]** | Indonesia | S | Peer education to improve menstrual knowledge and reduce anxiety |

## Inclusion criteria

The systematic review included randomised controlled trials (RCTs), observational studies, quasi-experimental studies and studies with a pre- and post-test design. All eligible studies had

to include at least one mental health or wellbeing outcome (either observational or self-report). The intervention under evaluation must have been at least partly peer-led; therefore, programs jointly led by a peer and an adult were eligible. Studies set in a primary, secondary or special education school, or further education institution for those under 18 years old, were included. School interventions that had an online delivery element were included only if a peer leader was involved. The format of the intervention could be either one-to-one or group-based, as long as any groups were at least partly peer-led. Any studies where an adult facilitated peer-to-peer contact, such as a teacher leading a discussion group, were not included.

We included studies that looked at either leader or recipient outcomes, or both. Studies were eligible even if they evaluated only the training component for a peer-led programme. Within our protocol, we specified that all peers had to be of school age (4–18 years old) and a current student within the intervention school. However, we expanded the age range to include slightly older students if a study was based in a country or culture where it was not uncommon to be at school beyond 18 years. Both quantitative and qualitative studies were included.

Studies including young people with or without a diagnosis of any psychological, emotional or behavioural conditions were eligible, so long as they attended school. We included studies with a minimum sample size of 50 peer pairs in the intervention group, or 50 peer leaders or recipients if only one group was reported.

### Risk of bias assessments

A comprehensive risk of bias assessment was carried out using validated and well-established assessment tools. TK and MF independently assessed each study in order to establish inter-rater reliability. All risk of bias was assessed using the Cochrane Risk of Bias Assessment Tool for randomised controlled trials [80], the Joanna Briggs Institute (JBI) Critical Appraisal Checklist for quasi-experimental studies [81], and the NIH Quality Assessment Tool for studies with a pre-post design [82]. The evaluations are included in S2 Appendix.

## Systematic review results

### Description of studies

A total of 45,597 studies were identified after the initial search and 11 were eligible for inclusion in the final analysis (see Fig 2 for flow diagram of search). The included studies had an estimated total sample size of 2,239 participants (929 peer leaders, 1,310 recipients) (see Table 2 for included studies). Of these, four studies used a randomised controlled design [20, 39, 71, 83]; six studies used a quasi-experimental design [5, 12, 66, 84–86]; and one operated a pre- and post-test design [87]. Three of the identified studies were unpublished dissertations [66, 84, 87]. Pupil sample sizes ranged from 55 [83] to 372 [12] and the number of experimental schools evaluated per study ranged from one [77] to 22 [36]. An unpublished dissertation [84] reported on several components of one study, however only the component examining the effect of the programme on the peer leaders is included here. One peer support programme, 'Natural Helpers', was included in two studies; both were doctoral dissertations on different samples [66, 87]. All studies evaluated interventions in secondary schools, with two also evaluating a primary school programme [11, 12]. Only one intervention was co-led with a teacher [84]. All other interventions were exclusively peer-delivered, although most included an element of adult supervision or guidance for the peer leaders. The age of peer leaders ranged from 12 to 18; recipients were aged between nine and 20 years and were usually younger than the peer leaders. Peer leaders were selected based on a range of criteria (shown in Fig 3). Interventions addressed mental health and wellbeing outcomes in the context of bullying [12],

**Table 2. Characteristics of included studies.**

| Author, year and location | Research design | N | Peer leader ages/ characteristics | Intervention type | Intervention characteristics | Frequency/ duration of program | Leader selection process | School type | Training | Leaders: Outcome measures and findings | Recipients: Outcome measures and findings |
|---|---|---|---|---|---|---|---|---|---|---|---|
| Bausano (2006), USA (Dissertation) | Pre-post | 110 mentors | Ages 13–18 | Peer support | 'Natural Helpers' program. Students identified as existing 'natural helpers' within the school given training to enhance skills. | ~ 6 months. Unstructured meetings | Anonymous peer nominations | Secondary | Practice basic counselling skills, e.g., validation of emotions, problem-solving skills. Emphasis on positive coping strategies. Where/when/ how to refer on. | Positive and Negative Affectivity Scale: no significant effect. | Not evaluated |
| Ellis (2004), Australia (Dissertation) | Quasi-experimental | 99 mentors | Ages 15–17 | Peer support | Peer leaders co-facilitate sessions with a supervising teacher. Each group contains 8–10 year 7 students and 2 peer leaders. Aims to help develop positive attitude and resilience. | 12 x 45-minute weekly peer support sessions for 12 weeks | School-set criteria and answers to questions | Secondary | Intensive 2-day training. Emphasis on: creating a supportive atmosphere; active listening; leadership skills. Focusing on identifying needs and empathising with issues, instruction giving, planning a session. | SDQII-S and ROPE: non-significant improvement to self-confidence | Not evaluated |
| **Ellis et al. (2009), Australia** | Quasi-experimental | 452 mentees | Mentees: ages 12–13; Mentors: ages 15–16 | Peer support | Peer leaders co-facilitate sessions with a supervising teacher. Each group contains 8–10 year 7 students and 2 peer leaders. Aims to help develop positive attitude and resilience. | 12 x 45-minute weekly peer support sessions for 12 weeks | School-set criteria and answers to questions | Secondary | Intensive 2-day training. Emphasis on: creating a supportive atmosphere; active listening; leadership skills. Focusing on identifying needs and empathising with issues, instruction giving, planning a session. | Not evaluated | SDQII-S and ROPE: no significant immediate effect on self-esteem. Significant f/u effect on self-confidence. |

(Continued)

**Table 2.** (Continued)

| Author, year and location | Research design | N | Peer leader ages/ characteristics | Intervention type | Intervention characteristics | Frequency/ duration of program | Leader selection process | School type | Training | Leaders: Outcome measures and findings | Recipients: Outcome measures and findings |
|---|---|---|---|---|---|---|---|---|---|---|---|
| Froh (2004), USA (Dissertation) | Quasi-experimental | 58 mentors | Ages 11–18 | Peer support | 'Natural Helpers' program. Students identified as existing 'natural helpers' within the school are given training to enhance helping skills. | Program running for > 4 years prior to study. Unstructured meetings | School-wide peer-nomination process | Secondary | 3-day retreat. Positive self-coping strategies. Effective interventions, e.g. facilitative listening. | Personal Development Inventory: non-significant increase in measure that includes self-confidence | Not evaluated |
| **Mason-Jones et al. (2013), South Africa** | RCT | 203 peer educators | ~16 years old | Sexual health and wellbeing education | N/A–training program for peer educators only. | N/A | Volunteers. Additional selection criteria unclear | Secondary | 2 x 1-hour skills sessions per month for duration of intervention. Delivered by non-profit organisations. Talk groups and mentoring sessions. 3-day camps–information sessions. Focused on developing range of psychosocial skills in the peer leaders | Self Esteem Questionnaire: non-significant improvements to self-esteem | Not evaluated |
| **Roach (2014), England** | Quasi-experimental | 372 mentees | Mentees: ages 9–12 Mentors: described as 'older' but specific ages not given | Mentoring & anti-bullying | Mentors aimed to establish supportive mentoring relationships. Drop-in service, one-to-one mentoring and group mentoring. | Ranged from 1–30 sessions | N/A | Primary and secondary | Training varied across schools | Not evaluated | Students Life Satisfaction Scale (modified): Mentees: higher levels of life satisfaction after 1 year. Girls scored lower than boys. Older mentees: more likely to have lower life satisfaction 1 year post intervention. Mentees who had fewer meetings recorded higher life satisfaction scores at 1-year follow-up. |

(*Continued*)

**Table 2.** (Continued)

| Author, year and location | Research design | N | Peer leader ages/ characteristics | Intervention type | Intervention characteristics | Frequency/ duration of program | Leader selection process | School type | Training | Leaders: Outcome measures and findings | Recipients: Outcome measures and findings |
|---|---|---|---|---|---|---|---|---|---|---|---|
| **Sebire et al. (2018), UK** | Cluster RCT | 249 supported peers | All participants female and aged 12–13 years old | Physical activity (PA) promotion | 'PLAN-A' intervention (Peer-Led physical Activity iNtervention). | 10-week period of peer leaders asked to informally diffuse messages about importance of PA to friends and support to maintain/ increase PA levels. | 'Influential' girls in year 8 nominated by same-year peers. Top 18% of nominations were invited to train as peer supporters. | Secondary | Initial 2-day course, followed by 1-day top-up around 5 weeks later. Training took place off-site at various locations, depending on school. Content involved education about importance of PA; empowerment; development of personal skills; different methods of support; core objectives of program. | Not evaluated | Health-related quality of life (EQ-5D-Y): very small effect size in intervention group between T1 (post-study) and T2 (follow-up) (Cohen's $d = 0.088$) |
| **Shah et al. (2001), Australia** | Cluster RCT | **Group 1:** 66 mentees **Group 2:** 47 mentees* (total = 113) | **Group 1:** Mean age 15.5 **Group 2:** Mean age 12.5 | Physical health education | 'Triple A' asthma management programme. Mentors used games, videos, worksheets and discussions to teach group 1. Group 1 gave interactive (short acts, drama and song) performances to group 2 on asthma management. | Mentors prepare 3x 45 minute lessons. Younger students prepare presentation | Students in older year group (age 16–17) volunteered as mentors for group 1; all group 1 students recruited to act as mentors to group 2 | Secondary | Workshop for older mentors (age 16–17) ('asthma peer leaders'), how to use interactive delivery methods to teach peers | Not evaluated | Paediatric Asthma Quality of Life Questionnaire: group 1 and 2 completed questionnaire. Non-significant improvement in overall quality of life in both groups. Males showed significant improvement ($p = 0.02$; 95% CI) in emotional domain. |
| **Song et al. (2018), China** | Quasi-experimental | 118 mentors; 124 mentees | Ages 12–14 | Academic tutoring | Students divided by gender then split into highest and lowest performing halves of class. Required to study together for 30 minutes per day, four times per week. ~ 1 semester. | Range | Highest performing students in class | Secondary | ? | 100-item 'Mental Health Questionnaire': sig. increase in levels of guilt; sig. decreased social stress | 100-item 'Mental Health Questionnaire': sig. decrease in mental health scores; sig. increase in 'learning stress' |

*(Continued)*

**Table 2.** (Continued)

| Author, year and location | Research design | N | Peer leader ages/ characteristics | Intervention type | Intervention characteristics | Frequency/ duration of program | Leader selection process | School type | Training | Leaders: Outcome measures and findings | Recipients: Outcome measures and findings |
|---|---|---|---|---|---|---|---|---|---|---|---|
| **Wyman et al. (2010), USA** | RCT | 268 mentors | Mean age = 15.7 | Suicide prevention | 'Sources of Strength' suicide prevention programme. Peers engaged in schoolwide messaging through presentations, public service announcements, videos, text messages and social network sites. Engaged trusted adults and encouraged friends to do same | Range as multiple tools employed | Peer and teacher nominations. Peers chosen to reflect diverse range of friendship groups | Secondary | Four hours. Led by certified trainers–focusing on coping skills and available resources. Interactive training. | Questions on recent suicidal ideation: decrease in suicidality over 3 months but not significant between conditions | Not evaluated |
| **Yogev and Ronen (1982), Israel** | Quasi-experimental | 73 mentors | 16 years old | Academic tutoring | 'Young Tutors' peer tutoring programme. Took place in schools and municipal youth clubs. Met 2 x per week during school hours. | ? | Offered as elective subject to freshman students (aged 16 years) | Secondary | 3 hours per week incorporated into class time. Lectures by school counsellors and invited specialists on adolescence and teaching methods. Included role-play, modelling and case analyses. | Self-Esteem Scale: significant increase in mentors' self-esteem vs. comparison group | Not evaluated |

'?' represents not enough information provided in the text. Version of 'peer leader' kept as originally stated in study.

*This study used a cascading approach, whereby one year group delivered an intervention to a younger year group (group 1), who then passed aspects of that on to an even younger group (group 2).

**Also included small number of Children and Young People's Community Organisations (CYPCOs).

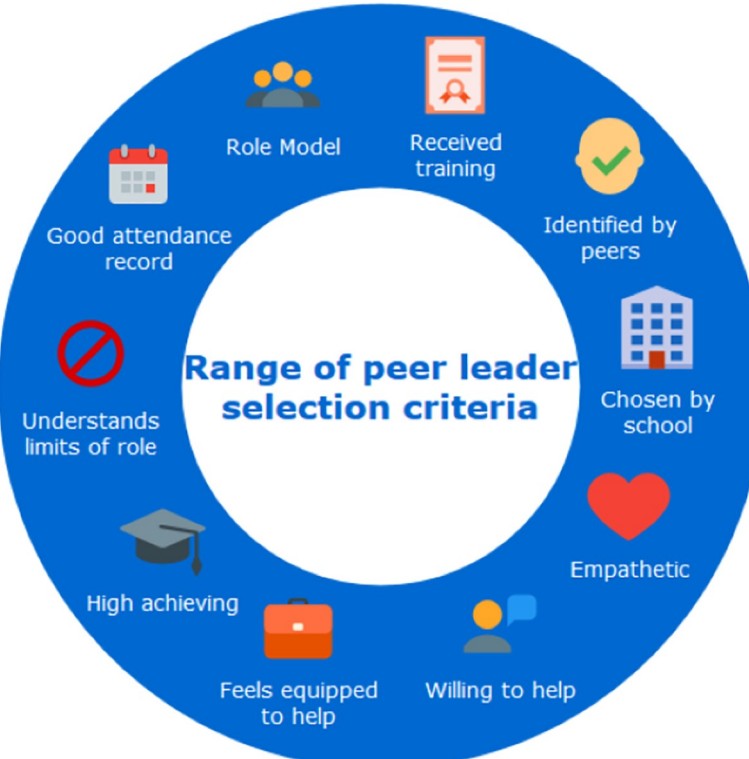

**Fig 3. Range of peer leader selection criteria seen across included studies.**

sexual health [71], physical health [20, 83], general psychological needs [5, 11, 66, 84, 87] and academic performance [85, 86]. Interventions were delivered through one-to-one sessions, group work, ad hoc drop-ins and school-wide 'messaging', e.g. disseminating information using posters, plays, announcements etc. One study was included where peers also worked in local municipal youth clubs but the majority of the intervention was conducted within a school [86]. It was not possible to perform a meta-analysis due to high levels of cross-study heterogeneity. It was therefore deemed most appropriate to conduct a narrative synthesis. In order to address the research question, the results are presented by outcome.

## Outcomes

**Peer leader outcomes.** *Self-esteem and self-confidence*. A study in Israel reported mixed findings, with a significant between-group effect ($F = 16.71$; $p < .001$) on self-esteem at the end of a paired tutoring programme [86]. Within-group changes were non-significant for peer leaders and a control group. The results also showed that the self-esteem of male tutors significantly increased compared to female tutors.

An RCT in South Africa measured the effects of training secondary school students to become peer leaders for a sexual health and wellbeing programme [71]. Self-esteem changes were compared between the group of students receiving the training (N = 203) and a control group (N = 302) who received none. They found no significant difference between the experimental and control groups after training.

Two studies evaluated the 'Natural Helpers' programme [66, 87], neither with significant findings, although one study reported a tendency towards improved self-confidence in peer leaders (M = 143.98, SD = 13.96) compared to a matched comparison group (M = 135.92,

SD = 15.28) [66]. This is similar to findings of a peer support scheme on the self-confidence of peer leaders aged 16 and 17 [84], where student leaders reported non-significant higher self-confidence scores than the control groups after the intervention and at follow-up.

*Positive and negative affectivity*. A quasi-experimental study measured positive and negative affectivity in peer leaders following a peer support programme [87]. No significant changes were found between groups of 'New Natural Helpers' (n = 54) and 'Experienced Natural Helpers' (n = 56) (those who were already in the role) when compared with two control groups (students who were nominated to be leaders but were not chosen by the research team (n = 51), and those who were not peer nominated (n = 61)).

*Social Stress*. A study in China found a significant decrease in mentors' 'social stress' scores (M = -0.497, SD = 0.209; $p<0.05$) following a one-to-one tutoring programme [85], but no significant change was observed in their 'overall mental health scores' (M = 64.32, SD = 13.85, treatment effect = -0.0952).

*Guilt*. Song et al. (see 'Social Stress' section) also measured levels of guilt experienced by peer leaders [85]. The peer leaders experienced significantly higher levels of guilt following the programme (M = 0.952, SD = 0.397; $p$ = 0.05), which the authors suggested might be due to their matched recipients not improving as much as they had hoped or because they had become more aware of inequalities amongst their peers.

*Anxiety*. Song et al. also included a brief, post-intervention survey evaluation, in which just over 42% of peer leaders reported that being a 'peer tutor' made them feel 'a little anxious', with 2.6% responding with 'very much' [85].

**Recipient outcomes.**   *Self-esteem & self-confidence*. An Australian study found that a peer support programme had no significant immediate or long-term effect on the self-esteem of students who received support from peer leaders [5]. A sub-group analysis of self-confidence in 12 and 13 year old students receiving the intervention showed that, although the intervention had no early effects, it had significant positive effects at follow-up (M .047, SD .017, $p <$ .01). These positive changes were also supported by qualitative findings [5].

*Suicidality*. An RCT sought to evaluate the whole-school effect of the 'Sources of Strength' suicide prevention programme in 18 high schools across the United States [39]. Students reporting 'some suicidal ideation' (over the previous 3 or 12 months) decreased in both the intervention (N = 268) and control (N = 185) groups over 3 months, with non-significant differences between conditions (*experimental*: pre: 14.8%, post: 11.6%; *control*: pre: 12.8%, post: 12.2%).

*Life satisfaction*. A study in England examined levels of life satisfaction in a sample of pupils (N = 372) who received an anti-bullying, peer-mentoring programme in primary (N = 6) and secondary (N = 16) schools [12]. Non-significant improvements to life satisfaction were seen in the student recipients compared to non-mentored students after one year, primarily in males. There were also non-significant results suggesting that recipients who attended a *low* number of meetings had higher levels of life satisfaction than those who took part in a 'medium' ($p$ = .14) or 'high' ($p$ = .74) number of meetings.

*Quality of Life*. A cluster RCT reported an improvement in overall quality of life for Australian secondary school students with self-reported asthma symptoms following a peer-led intervention [20]. A total of 113 students reporting a 'recent wheeze' received the intervention and completed all assessments. A significant improvement was reported for male recipients in the 'emotions' domain of the questionnaire (2.2% to 37.4%; 95% CI; $p$ = 0.02).

A separate RCT measured the health-related quality of life of female peer recipients following an intervention to increase levels of physical activity [83]. The results showed a very small effect size in the intervention group between post-study (end of year 8) and follow-up (beginning of year 9) (Cohen's $d$ = 0.088).

*Learning Stress.* A study in China (see 'Social Stress' section) found a significant increase in recipients' levels of 'learning stress' (1.027, SD 0.413; $p<0.05$) [85]. The authors suggest this may be due to increased daily study time or stress from wanting to improve performance. They also found that recipients' overall mental health worsened significantly over the course of the intervention (M = 61.45, SD 15.22; effect size = -4.115; p< 0.01).

*Anxiety.* Song et al. also included a brief, post-intervention survey evaluation in which around 43.5% of peer recipients reported that being a 'peer tutee' made them feel 'a little anxious', with almost 4.5% stating 'very much' [85].

## Training of peer leaders

There were a wide range of training approaches used across the studies, with certain common themes. Most focused on teaching the peer leaders basic counselling and psychosocial skills, such as active listening and creating a supportive, non-judgemental environment. Some also focused on helping the peer leaders to recognise when someone may need extra support and the correct referral channels to pursue in these circumstances. All training programmes in the systematic review included interactive elements, such as discussion groups, games, exercises, and role-play. The duration and intensity of training periods varied widely, ranging from one-off sessions lasting a few hours to multiple training sessions across the duration of the intervention period. None of the studies reported that the training they delivered was evidence-based. The evidence gathered in this review is not sufficient to determine the exact relationship between the nature of the training, e.g. duration, content or mode of delivery, and subsequent mental health outcomes.

## Role of peer leaders

The role of the peer leaders depended largely upon the aim of the intervention. Some interventions employed a universal health promotion strategy, for which peer leaders appeared to be useful as role models of positive behaviours and for spreading information, such as suicide awareness, amongst the student body. Other interventions provided support for those already experiencing a mental health difficulty, in which case the peer leaders generally took on the role of lay counsellors. While peer leaders took a central role in the delivery of these interventions, it is not clear from the detail included in the studies to what extent their involvement, or any other element of the intervention, had a direct effect on mental health outcomes. This is made particularly difficult by the lack of significant positive outcomes. In this sense, the *mechanisms* of peer-led interventions, i.e. the elements of change, are still unclear.

## Risks

The potential risks associated with these interventions are also not clear from the included studies. However, one study's findings present some concern over the potential iatrogenic effects of taking part in a peer-led intervention [85]. The study found a significant decrease in the overall mental health scores of peer recipients following a tutoring program. The authors suggested this may have been due to their classification as the lowest achieving in their class, as levels of 'learning stress' were also seen to increase significantly in this group. This may therefore reflect a response to possibly being singled out as under-achieving, which bears serious consideration both in a research context and in real-world application. In the same study a group of peer leaders, who were trained as tutors after having been identified as in the top half of the class, reported significantly higher feelings of guilt following the intervention. The authors suggest this may have developed from not feeling that their input helped their tutees succeed academically. This sense of responsibility over the outcomes of an intervention is a

major consideration when recruiting young people as primary delivery agents and suggests that combining the delivery between pupils and adults, e.g. teachers, might alleviate some of this responsibility. Beyond these findings, it is not possible to draw any firm conclusions around the potential risk of peer-led interventions due to a paucity of data in this area, making this a key research priority given the sensitive nature of these interventions and their placement within schools. Indeed, it seems highly important to consider the unique context of these interventions and if, for example, there is transference of confidential information between school peers, and the complex confidentiality and safeguarding issues this could pose.

## Discussion

This review has evaluated the evidence for peer-led mental health interventions in schools. The scoping review of 54 studies highlighted the various uses of these interventions and that student peers have tried to play both supportive and educational roles using a range of intervention designs, for example from facilitating peer connections to suicide prevention. However, when examining efficacy of peer-led school-based interventions in the systematic review, only 11 studies were identified as eligible for inclusion; seven explored peer leader outcomes, five explored peer recipient outcomes (one study looked at both). Most studies were assessed as at low risk of bias. The majority of studies were conducted in high-income nations.

In the seven studies reporting on peer-leader effects, only two [85, 86] reported significant findings—these were on improved self-esteem, decreased social stress and increased guilt. Of note, both these studies were of academic tutoring programmes. In the five studies reporting on peer-recipient outcomes, two studies [5, 20] reported positive significant findings on self-confidence and a measure of quality of life; and one study [85] reported on negative impacts on learning stress and general mental health.

The findings from the included studies are therefore not adequate to make any firm conclusions about the effectiveness of these interventions, especially given the small number of significant results and the heterogeneity of measured outcomes which is of some concern given their seemingly widespread application in many school systems. All reported outcomes were related to wellbeing issues and emotional difficulties, with no studies measuring a diagnosable psychiatric disorder, such as a depressive or anxiety disorder.

Based on the findings of this review, it is clear that further research using rigorous scientific methodology is needed. Only three RCTs met our eligibility criteria and the sample sizes across the reviewed studies varied. Further research to explore the potential risks of these interventions would also need to be considered, as one study reported multiple significant negative effects of taking part in a tutoring intervention and determining if this was because of the school environment, the intervention, or cultural factors needs to be explored.

If peer-led interventions are developed, a focus on establishing best practices for key elements such as peer selection, training and supervision, and delivery would be important. Little rationale was given across the studies for these key design decisions and little evidence exists in the wider literature as a guide. Furthermore, most studies included in the systematic review failed to describe core elements of their implementation procedures or conduct any evaluation of them. The majority of studies chose peer leaders based on a set of criteria developed either by the school or the research team, and predominantly the teachers would make the selection. In instances where peer nominations were involved, the final selection was typically still made by the school staff. None of the studies in the systematic review included any qualitative data on young people's perceptions of what a 'peer' is, although this would provide essential insight into what would be acceptable to pupils. Future research could compare outcomes of

interventions using solely teacher-selected, partly pupil- and teacher-selected, and solely pupil-selected peers, as it is unclear which method of selection is most effective.

A large UK report of an initiative to encourage schools to promote and run peer-led interventions in 89 primary, secondary, and special education schools, as well as in community groups, complement these findings [11]. Each school designed and implemented their own peer support programme based on guidelines provided by a national delivery partner, who also ran trainer-training sessions. It was at the discretion of schools to determine the most appropriate methods for the recruitment, training and supervision of peer mentors, matching arrangements with mentees, and the mode, frequency and duration of each local intervention. The study therefore measured a heterogeneous group of peer-led interventions, some of which might have fulfilled our inclusion criteria, but it was not possible to obtain disaggregated data. A pre-post design was used to measure a range of mental health and well-being outcomes across peer leaders, recipients, young people who performed both roles within the programme, and non-participants. Of the different measures collected, the only statistically significant improvement observed was in a Community Connection sub-scale of a resilience measure, at both 3 and 9 month follow-up, in primary aged children (n = 373) who could have been either peer leaders, recipients or neither. Qualitative data collected from young participants involved in the interventions seemed to indicate that peer leaders were generally more positive about the programme, although the recipients interviewed often wanted to have been able to spend more time with their matched peer. There was also some emphasis on the importance of young people leading the implementation of the programmes to ensure success.

This review has a number of strengths. Firstly, the systematic review is the first of its kind to isolate the mental health effects of school-based, peer-led programmes, which is important given their widespread use. Our literature search was extensive, covering 11 academic databases and thus increasing the likelihood of all relevant studies being captured. Secondly, by also conducting a scoping review we were able to provide a novel and timely map of the many ways in which these interventions have been used. This review has sought to address a current and urgent research gap in an area of national interest, substantial activity and investment in time. It hopes to inform the evaluation of these programmes going forward.

## Limitations

This review has several limitations. Firstly, although every attempt was made to capture as many mental health and wellbeing outcomes within the search as possible, the language around mental health is broad and fluid which means we may have missed certain terms. However, we have made an example of our search terms available for replication or adaptation (S1 Appendix). Secondly, the number of results returned from the initial literature search suggest that the search strategy may need to be refined. However, we chose to ensure a large return given the broad nature of the subject, poor reporting of outcomes in abstracts and that the search was applied to a large number of databases. Studies of broader types of peer support might have been missed if they included mental health outcomes amongst a number of other reported areas. Thirdly, a very small number of search terms that appeared after 'exploding' certain terms during the initial literature search were not available when we decided to repeat the search at a later date. Where an exact term was not available, it was omitted from the search. However, these were generally highly specific terms that are unlikely to have caused key texts to be missed. Lastly, the exclusion of studies with smaller sample sizes may have also been a limitation. Of the studies excluded based on sample size, the majority had very small samples, however we sought to summarise any study excluded based on sample size that were

closer to our inclusion threshold. This process identified four studies (in Canada, Ireland, England and Malaysia) that had slightly larger sample sizes (range of $N = 27–46$) but were insufficient for inclusion [53, 59, 64,'68]. One study found a significant reduction in test anxiety in an adolescent group exposed to 'peer coaching'; however, none of the remaining studies reported any significant mental health or wellbeing outcomes.

## Conclusion

Given their seemingly widespread use in schools, peer-led mental health interventions need to be better assessed and their impacts understood so as to ensure that if used, they can target those children most likely to benefit. Although young people are a potentially important resource to provide low-intensity mental health support in school settings, the current evidence base does not support widespread implementation and therefore further evaluation of existing programmes needs to be prioritised. Despite peer-led mental health interventions being often developed to help peer recipients, the data shows that the peer leaders can also benefit. It might be that with the training and supervision often provided to peer leaders, along with a possible improvement in their self-esteem, that more vulnerable children should be asked to be peer leaders and not just recipients. Understanding the results and the future design of interventions would probably benefit most from the direct input of young people who are well-placed to co-design such intervention decisions [88].

## Supporting information

**S1 Checklist.**
(DOC)

**S1 Appendix. Example systematic review search (PsycINFO).**
(DOCX)

**S2 Appendix. Risk of bias assessments.**
(DOCX)

## Acknowledgments

The authors would like to thank Laurie Day, Diarmid Campbell-Jack and Professor Yang Song for providing further information about their research, as well as Julia Hallam for helping to develop the search strategy.

## Author Contributions

**Conceptualization:** Thomas King, Mina Fazel.

**Data curation:** Thomas King, Mina Fazel.

**Formal analysis:** Thomas King, Mina Fazel.

**Methodology:** Thomas King, Mina Fazel.

**Supervision:** Mina Fazel.

**Writing – original draft:** Thomas King.

**Writing – review & editing:** Thomas King, Mina Fazel.

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
