## [Decision Letter · Decision Letter 0]

18 Nov 2020

PONE-D-20-27585

Examining the mental health outcomes of school-based peer-led interventions on young people: a scoping review of range and a systematic review of effectiveness

PLOS ONE

Dear Dr. King,

Thank you for submitting your manuscript to PLOS ONE. After careful consideration, we feel that it has merit but does not fully meet PLOS ONE’s publication criteria as it currently stands. Therefore, we invite you to submit a revised version of the manuscript that addresses the points raised during the review process.

We look forward to receiving your revised manuscript.

Kind regards,

Danuta Wasserman

Academic Editor

PLOS ONE

Journal Requirements:

Reviewers' comments:

Reviewer's Responses to Questions

**Comments to the Author**

1. Is the manuscript technically sound, and do the data support the conclusions?

Reviewer #1: Partly

Reviewer #2: Partly

2. Has the statistical analysis been performed appropriately and rigorously? 

Reviewer #1: N/A

Reviewer #2: Yes

3. Have the authors made all data underlying the findings in their manuscript fully available?

Reviewer #1: No

Reviewer #2: Yes

4. Is the manuscript presented in an intelligible fashion and written in standard English?

Reviewer #1: No

Reviewer #2: Yes

5. Review Comments to the Author

Reviewer #1: The manuscript reports a review on peer-led school-based intervention for mental health. The topic is important and the authors are recommended for doing research in this field. The manuscript is also well written and the summary of studies included in the review is informative. There are some issues that warrant further attention, in particular regarding the methodology of this review. These are outlined below.

The introduction is a bit lengthy and not well structured. Methodological descriptions alternate with background information and arguments for peer-led interventions.

I missed a clear description of the search strategy, which in my opinion is mandatory. How exactly did the authors search for the literature? At this stage, it is not possible to judge whether this review is comprehensive but this is one of the major quality criteria for a systematic and also for a scoping review. E.g. the definition “mental health outcome” is very broad. How exactly did the authors search for this and make sure that they did not miss many of those?

Who did screening, data extraction, ect.? Was this done by one or several persons? And was there double coding or at least some quality control?

I think that the combination of the scoping review and the systematic review is a questionable choice, in particular since the two seem poorly integrated and are described in two separate method and results sections. I accept that the authors aim to integrate both into one paper; however, I feel that more efforts are needed to really “integrate” them. As an alternative, I feel that the scoping review is not necessary here.

Reviewer #2: This review missed the mark for me. So much time and effort went into the review but the results and discussion did not make major points about the science base needed to move the field forward. The iatrogenic impacts should be more clear as schools should use this in their decision-making. Programs with known iatrogenic impact should be avoided.

6. PLOS authors have the option to publish the peer review history of their article (what does this mean?). If published, this will include your full peer review and any attached files.

Reviewer #1: No

Reviewer #2: No

---

## [Author Response · Author response to Decision Letter 0]

26 Jan 2021

Reviewer #3

The focus of this review is on the range and effectiveness of peer led programs. Many schools use peer approaches informally to help with transitions, i.e., assign a “buddy” to help a new student acclimating to a new school. This makes practical sense and can bridge a stressful transition and build a sense of social connectedness which is important to student mental health in general. 

Peers may be in a better position to identify students struggling with mental health problems and other challenges than adults in the schools. It makes practical sense to provide some mental health education to all students in schools so they are able to identify a student in crisis and know what to do to facilitate that student being connected to an adult who can get help them the help they need. There are established effective models for this (e.g., Teen Mental Health First Aid). 

1. Clarify the overall results:

It was impressive that 11 databases were used (very comprehensive) but, to me, this review missed the mark. A great deal of time and effort went into this review but it is not clear to me which programs are effective and can be scaled up. Maybe that was because the studies are not adequate to really say anything conclusive about the range and effectiveness of peer led programs. If there are not any, please be explicit about this.

DONE: Thank you, we are sorry that this was not clear and we have made a number of changes throughout the manuscript to make sure that the fact that the evidence-base is seriously lacking will be highlighted further. We used 11 databases as we wanted to make sure that as many possible sources of publication were captured as these interventions can be published in education, social care, health, mental health and sociology journals. We were also surprised that given how extensive our search was and how common peer-led interventions are, there is little evidence to support their use. This is why we are so keen to get this review to a broader audience as there is an assumption, we believe, that such interventions must have a stronger evidence-base to support their widespread use. 

2. Training and role of peer leaders

It would be great if you could better synthesize/summarize the details for the training provided for peer leaders – how many hours, what is the training content, were adults involved? In an ideal world, peer leaders should be trained to inform a trusted adult in the school about students who are at risk. The role of the peer leaders should also be more explicit, please synthesize and summarize this as well. It would have been ideal for the authors to describe the role the peers played and elements of the interventions that have a positive impact on outcomes. Importantly, were the peer leaders connected with adults or supervised by adults? The Sources of Strength program is very good but I believe 10% off the staff are trained in addition to the students and there are adults that supervise the student leaders and support them. This is a robust model that has been around for decades and has been broadly disseminated. Maybe this should be outlined in the results and discussion sections with subheadings (Training of peer, role of peer, school adult involvement). 

 DONE: Thank you for raising this important point. We have documented the training that the peer-leaders have had in the summary table and have now also summarised this in the results section. Our concern in focusing too much on this component is that although there were more positive results for peer-leaders than for the peers identified to be mentored, there might then be an assumption that if more peers are trained, then this will have beneficial results for peer recipients, which is not necessarily the case. We believe that for the programmes where there are positive mental health outcomes for the peer-recipients then focus should be placed on the training of their peer leaders. But we are reluctant to highlight training which does not result in measured positive mental health impacts on recipients. 

3. Active components of programmes

Which components or elements of programs appear to be effective and, more importantly, which appear to be harmful? One of the most important take-aways for me is the authors pointing out the risk of harm to students or iatrogenic effects. These should be clearly outlined, this point should be explicit so that schools can use this in decision-making and either prepare for this to avoid these outcomes or decide to select an adult-led program. Guilt was mentioned as an outcome. Did peer leaders feel guilty because they were not able to prevent bullying or suicide, and that is why they feel guilty? If so, this is a major issue that should be elucidated for schools. 

CONSIDERED: thank you for making this point, we have tried to highlight these components further but in reality the number of studies is too small to make such conclusions and we believe this is the main point of the review, we do not think we can make too many other conclusions and give detailed evidence of which components might be harmful, for example, as the data does not lent itself to more than just a narrative review at this stage. 

4. Evidence for adult-led programmes

Given that there are several adult-led school-based programs that are effective (e.g, Youth Aware of Mental Health, Good Behavior Game, Teen Mental Health Frist Aid) , perhaps the recommendation should be that these effective programs should be prioritized over the peer-led programs until a stronger evidence base is established for peer programs. 

CONSIDERED: thank you for making this point but our review only examined peer-led interventions and so we are restricting our discussion to this area. We have not conducted a similar review of adult-led interventions although are reassured that this is an area where more studies have taken place. We believe at this stage, greater discussion with young people might best highlight the most acceptable modes of interventions and have added this to the conclusion, raising the point made above.

5. Overall conclusions

It does not seem that these programs prevented bullying, suicidal behaviors or the outcomes they targeted but they appear to make students feel more connected to one another and increase self-esteem or the peer leaders. Perhaps the follow-up of the studies was not long enough to see impacts. Perhaps the samples size was too small? Perhaps the schools did not properly measure these outcomes. The authors need to be more explicit about what is needed to move the science forward in this area. As it appears that is 62% of schools in England use these programs, there needs to be a push for evaluation of the efforts. 

Also, the sustainability of these programs is also of interest. Are there any data on these programs being sustained long-term? 

DONE: thank you these are all important points and we have added them to the discussion section. None of the studies conducted longer-term follow-up, to our knowledge. 

Reviewer #1: 

The manuscript reports a review on peer-led school-based intervention for mental health. The topic is important and the authors are recommended for doing research in this field. The manuscript is also well written and the summary of studies included in the review is informative. There are some issues that warrant further attention, in particular regarding the methodology of this review. These are outlined below.

1. More focused introduction needed

The introduction is a bit lengthy and not well structured. Methodological descriptions alternate with background information and arguments for peer-led interventions.

DONE: thank you, we have tried to restructure and refocus the introduction. 

2. Search strategy

I missed a clear description of the search strategy, which in my opinion is mandatory. How exactly did the authors search for the literature? At this stage, it is not possible to judge whether this review is comprehensive but this is one of the major quality criteria for a systematic and also for a scoping review. E.g. the definition “mental health outcome” is very broad. How exactly did the authors search for this and make sure that they did not miss many of those?

DONE: we published the protocol for this systematic review where these methodological points are described (King, T., Fazel, M. Examining the mental health outcomes of peer-led school-based interventions on young people aged between 4 and 18 years old: a systematic review protocol. Syst Rev 8, 104 (2019). https://doi.org/10.1186/s13643-019-1027-3) and we included the full search strategy as an appendix in this review manuscript. 

3. Screening process

Who did screening, data extraction, ect.? Was this done by one or several persons? And was there double coding or at least some quality control?

DONE: we have now added this clarification

4. Integration of the scoping and systematic review

I think that the combination of the scoping review and the systematic review is a questionable choice, in particular since the two seem poorly integrated and are described in two separate method and results sections. I accept that the authors aim to integrate both into one paper; however, I feel that more efforts are needed to really “integrate” them. As an alternative, I feel that the scoping review is not necessary here.

DONE: we are sorry that this has not been integrated. We have tried to address this by referring to the rationale adopted in the introduction to the review and then also in the results and the discussion. We believe both aspects are important as there are so many different types of peer-led interventions taking place in schools yet as there is such a negligible evidence-base we wanted to find a way to describe an map the field and then also do the same for the evidence. This way a practitioner or researcher will have a better appreciation of the field as a whole. Ideally there would have been no need for the scoping review as the studies included in the systematic review would have also demonstrated the range of work conducted in this area. However, given that the number of studies was so small, we believe both are important. In order to try and address this, we have done two things, we have integrated the areas more and also placed less emphasis on the scoping review findings in the results section and hope that sufficient information will be evident from the table. 

Reviewer #2: 

This review missed the mark for me. 

1. Focus of the discussion:

So much time and effort went into the review but the results and discussion did not make major points about the science base needed to move the field forward. 

DONE: we have tried to make our discussion more focused to clarify the existing evidence-base as well as direct future scientific enquiry so as to help move the field forward which we agree, is desperately needed. For example, in the discussion we now state:

‘Based on the findings of this review, it is clear that further research using rigorous scientific methodology is needed. Only three RCTs met our eligibility criteria so future studies should look to use this gold-standard methodology where possible. The sample sizes across the reviewed studies varied and a number of studies in our initial search were excluded due to very small sample sizes, therefore future studies should focus on evaluating powered samples where possible. Much more research is also required to explore the potential risks of these interventions in further detail, as one study saw significant negative effects of taking part in an intervention… Future research should also aim to build the evidence base for these interventions from the ground up, with a focus on establishing best practices for key elements such as peer selection, training and supervision, and delivery.’ 

2. More emphasis on iatrogenic effects

The iatrogenic impacts should be more clear as schools should use this in their decision-making. Programs with known iatrogenic impact should be avoided.

DONE: we agree that these effects are concerning and we have tried to draw more attention to them in the results section and discussion. We are also aware that the number of studies is small and participants are also small in number and want to highlight the overall limitations of the evidence-base.

---

## [Decision Letter · Decision Letter 1]

22 Mar 2021

Examining the mental health outcomes of school-based peer-led interventions on young people: a scoping review of range and a systematic review of effectiveness

PONE-D-20-27585R1

Dear Dr. King,

We’re pleased to inform you that your manuscript has been judged scientifically suitable for publication and will be formally accepted for publication once it meets all outstanding technical requirements.

Kind regards,

Danuta Wasserman

Academic Editor

PLOS ONE

Additional Editor Comments (optional):

Reviewers' comments:

Reviewer's Responses to Questions

**Comments to the Author**

1. If the authors have adequately addressed your comments raised in a previous round of review and you feel that this manuscript is now acceptable for publication, you may indicate that here to bypass the “Comments to the Author” section, enter your conflict of interest statement in the “Confidential to Editor” section, and submit your "Accept" recommendation.

Reviewer #1: All comments have been addressed

2. Is the manuscript technically sound, and do the data support the conclusions?

Reviewer #1: Yes

3. Has the statistical analysis been performed appropriately and rigorously? 

Reviewer #1: N/A

4. Have the authors made all data underlying the findings in their manuscript fully available?

Reviewer #1: Yes

5. Is the manuscript presented in an intelligible fashion and written in standard English?

Reviewer #1: Yes

6. Review Comments to the Author

Reviewer #1: The authors have sufficiently addressed my comments.

7. PLOS authors have the option to publish the peer review history of their article (what does this mean?). If published, this will include your full peer review and any attached files.

Reviewer #1: No

---

## [Editor Report · Acceptance letter]

5 Apr 2021

PONE-D-20-27585R1 

Examining the mental health outcomes of school-based peer-led interventions on young people: a scoping review of range and a systematic review of effectiveness 

Dear Dr. King:

I'm pleased to inform you that your manuscript has been deemed suitable for publication in PLOS ONE. Congratulations! Your manuscript is now with our production department. 

Kind regards, 

on behalf of

Dr. Danuta Wasserman 

Academic Editor

PLOS ONE